# Crystal Structure of the Cyclostreptin-Tubulin Adduct: Implications for Tubulin Activation by Taxane-Site Ligands

**DOI:** 10.3390/ijms20061392

**Published:** 2019-03-20

**Authors:** Francisco de Asís Balaguer, Tobias Mühlethaler, Juan Estévez-Gallego, Enrique Calvo, Juan Francisco Giménez-Abián, April L. Risinger, Erik J. Sorensen, Christopher D. Vanderwal, Karl-Heinz Altmann, Susan L. Mooberry, Michel O. Steinmetz, María Ángela Oliva, Andrea E. Prota, J. Fernando Díaz

**Affiliations:** 1Structural and Chemical Biology Department. Centro de Investigaciones Biológicas, CSIC, Ramiro de Maeztu 9, 28040 Madrid, Spain; pacobal@cib.csic.es (F.d.A.B.); jeg@cib.csic.es (J.E.-G.); gimenezjf@cib.csic.es (J.F.G.-A.); marian@cib.csic.es (M.Á.O.); 2Laboratory of Biomolecular Research, Division of Biology and Chemistry, Paul Scherrer Institut, 5232 Villigen PSI, Switzerland; tobias.muehlethaler@psi.ch (T.M.); michel.steinmetz@psi.ch (M.O.S.); 3Unidad de Proteómica. Centro Nacional de Investigaciones Cardiovasculares, CNIC. Melchor Fernández de Almagro 3, 28029 Madrid, Spain; ecalvo@cnic.es; 4Department of Pharmacology, The University of Texas Health Science Center at San Antonio, San Antonio, TX 78229-3900, USA; risingera@uthscsa.edu (A.L.R.); mooberry@uthscsa.edu (S.L.M.); 5Department of Chemistry, Princeton University, Princeton, NJ 08544, USA; ejs@princeton.edu; 6Department of Chemistry, 1102 Natural Sciences II, University of California, Irvine, CA 92697-2025, USA; cdv@uci.edu; 7ETH Zürich, Department of Chemistry and Applied Biosciences, Institute of Pharmaceutical Sciences, 8093 Zürich, Switzerland; karl-heinz.altmann@pharma.ethz.ch; 8University of Basel, Biozentrum, 4056 Basel, Switzerland

**Keywords:** cyclostreptin, tubulin, microtubules, multidrug resistance, taxanes

## Abstract

It has been proposed that one of the mechanisms of taxane-site ligand-mediated tubulin activation is modulation of the structure of a switch element (the M-loop) from a disordered form in dimeric tubulin to a folded helical structure in microtubules. Here, we used covalent taxane-site ligands, including cyclostreptin, to gain further insight into this mechanism. The crystal structure of cyclostreptin-bound tubulin reveals covalent binding to βHis229, but no stabilization of the M-loop. The capacity of cyclostreptin to induce microtubule assembly compared to other covalent taxane-site agents demonstrates that the induction of tubulin assembly is not strictly dependent on M-loop stabilization. We further demonstrate that most covalent taxane-site ligands are able to partially overcome drug resistance mediated by βIII-tubulin (βIII) overexpression in HeLa cells, and compare their activities to pironetin, an interfacial covalent inhibitor of tubulin assembly that displays invariant growth inhibition in these cells. Our findings suggest a relationship between a diminished interaction of taxane-site ligands with βIII-tubulin and βIII tubulin-mediated drug resistance. This supports the idea that overexpression of βIII increases microtubule dynamicity by counteracting the enhanced microtubule stability promoted by covalent taxane-site binding ligands.

## 1. Introduction

Microtubule-based chemotherapy is one of the most effective treatment options for both solid tumors and hematological malignancies [1,2] and continues to be used even in combination with new molecularly targeted approaches, including immunotherapies [3]. The identification of several classes of natural products that bind tubulin/microtubules has informed on multiple, discrete binding sites that can affect microtubule structure and dynamicity. These include microtubule destabilizing sites characterized by the binding of maytansine, the vinca alkaloids, eribulin, colchicine, and pironetin. In contrast, binding to the taxane-, the laulimalide-, or in the case of triazolopyrimidines, the vinca-site results in net microtubule stabilization. To date, only three of these sites are targeted by cancer chemotherapies: the maytansine (using maytansine as an antibody-drug conjugate (ADC)), the vinca (vinca alkaloids, eribulin, and monomethylauristatin E as a part of an ADC) [4] and the taxane (paclitaxel, docetaxel, cabazitaxel, and ixabepilone) site [5].

The successful use of maytansine and auristatin loaded ADCs indicates that these potent cytotoxic agents are extremely effective anticancer drugs when used as the warheads of tumor-targeting antibodies. It could be speculated that compounds targeting the taxane-site delivered with an antibody could similarly be a promising ADC strategy with a higher therapeutic index. However, the development of ADCs necessitates compounds with an extremely high affinity or a covalent mode of action that can effectively engage the target, in spite of the lower concentrations delivered to the tumor by this method. Additionally, compounds that bind tubulin covalently are not susceptible to efflux by P-glycoprotein or other membrane efflux pumps due to the covalent nature of the interaction, thus escaping a major clinically relevant mechanism of resistance to chemotherapy [6,7,8].

Four classes of compounds that covalently bind to tubulin to modulate microtubule polymerization have been described (Figure 1). Pironetin perturbs key secondary structural elements at the interdimer interface by covalently binding to αCys316 (note that the tubulin residue numbering of [9] is employed in this manuscript), thereby preventing tubulin polymerization [10,11]. The other three classes of compounds bind to the taxane-site and function as microtubule stabilizing agents. Taccalonolides AF and AJ bind to βAsp226 [12], while zampanolide binds to βHis229 [8]. The fourth covalent tubulin binding compound is cyclostreptin [13], a weak tubulin polymerization enhancer, which has been described to covalently interact with tubulin at βThr220 and βAsn228 [6]. βThr220 is located at the pore site 1, which has been postulated to facilitate the entry of taxane-site ligands to their luminal binding site. Little is known about the structural mode of action of cyclostreptin.

Recent studies of microtubules and tubulin-bound taxane-site targeting agents [14,15] have shed light on the structural mechanisms of action of taxane-site ligands, but the matter is far from being fully understood. Epothilone A and zampanolide promote lateral protofilament association by stabilizing the S7-H9 loop (M-loop), a key secondary structural element establishing lateral contacts between adjacent protofilaments [14]. Paclitaxel-induced microtubule stabilization, which compensates for the contraction of the lattice resulting from the nucleotide hydrolysis, acts by stabilizing longitudinal interfaces to straighten protofilaments [15]. It has been described that epothilones, dictyostatins, zampanolides, and discodermolides all connect the base of the M-loop with β-tubulin helices H6 and H7, while paclitaxel would achieve a similar effect by connecting helices H1 and H7 with the M-loop [16,17]. In the case of taccalonolide AJ, the high-resolution crystallographic data [12] show the M-loop in an unordered conformation; likewise hydrogen-deuterium exchange experiments [18] indicate that the taccalonolides do not promote M-loop stabilization. Hence, it is likely that different microtubule-stabilizing agents (MSA) targeting the taxane-site have distinct mechanisms of action.

In the work described in this manuscript we have employed a multidisciplinary approach involving biochemical techniques and structural biology to understand how tubulin assembles into stable microtubules as a result of the binding of small molecules to the taxane-site. We utilize the different biochemical and structural properties of the covalent microtubule binders zampanolide, taccalonolide AJ, and cyclostreptin to reveal the structural basis for drug-induced microtubule assembly.

## 2. Results

### 2.1. Cyclostreptin Covalently Binds to the Taxane-Site and Does Not Induce M-Loop Folding

To obtain the structure of cyclostreptin bound to the two previously described binding sites (the pore site (binding at βThr220) and the taxane-site (binding at βAsn228) [6,19]), we prepared adducts under different conditions. First, using conditions in which the adduct at βThr220 should be predominant, the reaction was complete after 4 h at 25 °C, according to HPLC analysis (Appendix A). The integrity of the protein after the treatment was further confirmed by the ability to assemble into microtubules (Appendix A). In a second experiment, microtubules were prepared under conditions, where the principal reaction should occur at βAsn228 [6]. Those microtubules were subsequently depolymerized yielding tubulin dimers bound to cyclostreptin. The tubulin-cyclostreptin adducts obtained from the two experiments were finally assembled into T_2_R-TTL complexes to perform the crystallization experiments as described in the “Methods” section. The crystal structures were solved to 1.9Å resolution by X-ray crystallography and the T_2_R-TTL-cyclostreptin complexes superimposed well with the one obtained in the absence of a ligand (PDB ID 4I55, rmsd_overall_ of 0.415Å over 2062 C_α_-atoms), thereby suggesting that cyclostreptin-binding does not affect the overall conformation of tubulin (Figure 2A). Surprisingly, both of the above approaches for the preparation of covalent tubulin-cyclostreptin adducts led to crystals featuring positive electron density at the taxane-sites on both β-tubulin subunits (Figure 2A). We found cyclostreptin covalently linked to βHis229 in both cases, while no interaction with βAsn228 or βThr220 was detected. These findings demonstrate that cyclostreptin attaches to the taxane-site exclusively by reaction with βHis229 (Figure 2B,C). The non-covalent interactions with the binding site do not include any hydrogen bonds or salt bridges, and comprise hydrophobic contacts between the methyl groups C7, C22, and C28, and Asp226, Leu217, and Ala233, and between core ring E (Figure 1) with Leu371, respectively (Figure 2B,C).

In order to understand the discrepancy between these crystal structures and previous MS/HPLC results, we performed a higher precision MS analysis (Figure 3). We found that βHis229 is modified by cyclostreptin in the β-tubulin-derived tryptic peptide containing the sequence β^219^LTTPTYGDLNHLVSATMSGVTTCLR^243^. This histidine residue is contiguous with the previously characterized cyclostreptin interacting residue βAsn228 [6,19]. The proximity between the two residues, in combination with the low resolution provided by the hybrid triple quadrupole measurement that was used to analyze the cyclostreptin adduct previously, likely resulted in an error in the assignment of the ions since only signals with very low intensity could be used to determine the site of modification [6,19]. High resolution of the orbital traps greatly increases the sensitivity and specificity of the characterization. Thus, the greater accessibility of βHis229 and its crucial reactive role with other taxane-site ligands strongly suggests that the previous assignment of the reaction site to βAsn228 was erroneous and that the residue mediating cyclostreptin binding to tubulin is βHis229.

Although the final structure showed an overall folding essentially identical to the apo-state (Figure 2B), the βT5 loop was found in a GTP-bound conformation (Figure 2B) [20] in spite of the fact that the nucleotide found in the nucleotide pocket is GDP and not GTP. Cyclostreptin (Figure 1) is characterized by a highly strained core structure, leading to high reactivity of the double bond between C2–C17. Both the acylation of a nucleophile through attack on C1 or the addition of a nucleophile to C17 would be reasonable modes of covalent bond formation, as they would both lead to the release of strain created by the presence of the C2–C17 bridgehead double bond. The electron density related to cyclostreptin allowed us to unequivocally establish that the reaction of cyclostreptin with βHis229 occurs at C17 of the ligand, i.e., by 1, 4 addition of Nτ of the histidine side chain to the enolate double bond. (Figure 2D). This finding correlates with the strict need of the C2–C17 double bond for activity.

### 2.2. Implications of M-Loop Structuring by Taxane-Site Ligands

Taxane-site ligands have been proposed to induce microtubule assembly by driving M-loop folding into a helical structure observed in the assembled form [21]. The energy required to structure the M-loop in the assembly process is provided through the binding of the ligand, which makes the assembly reaction thermodynamically more favorable [14]. βHis229 is a key residue for the interaction of drugs with the taxane-site as described in several structural studies by X-ray crystallography, NMR and cryo-electron microscopy, which comprise complexes of tubulin with zampanolide and epothilone [14], dictyostatin [16], discodermolide (DDM), and the DDM-paclitaxel-hybrid KS-1-199-32 [17], taccalonolide AJ [12], docetaxel [22], and paclitaxel [15]; it is therefore an ideal candidate to be the reactive residue.

To gain further insight into the detailed mechanism that controls M-loop structuring, we superimposed all available crystal structures of taxane-site ligand-tubulin complexes and compared the ligand environment in the binding site. Based on the observed structural features, these taxane-site ligands can be subdivided into two groups, those that induce helical M-loop structuring, and those that show little to no effect on M-loop conformation (Figure 4). All ligands show common hydrophobic interaction points with helices βH6, βH7 and the M-loop base to different extents, which comprise βLeu217, βLeu219, βLeu230, βAla233, and βLeu275, and both the reactive side chains of βAsp226 and βHis229.

The first group of compounds, which comprises zampanolide (PDB ID 4I4T), epothilone A (PDB ID 4I50) and the DDM-paclitaxel hybrid KS-1-199-32 (PDB ID 5LXS), establishes extensive interactions with the M-loop through their corresponding sidechains, resulting in the induction of helical structuring (Figure 4A). The second group, which comprises discodermolide (DDM, PDB ID 5LXT) [17], dictyostatin (PDB ID 4MF4) [16] and taccalonolide AJ (PDB ID 5EZY) [12], features additional interactions with βArg278 and βGln282 but promotes only partial M-loop structuring with no detectable helical conformation (Figure 4B). This leads to the hypothesis that helical structuring requires more extensive interactions as provided by the side chains of group I compounds, and is further supported by the taccalonolide AJ-bound β-tubulin structure, where the observed disordered M-loop conformation is in agreement with hydrogen-deuterium exchange experiments [18] that indicate no structuration of the M-loop.

The covalent binding of cyclostreptin to βHis229 allows establishing comparable, but less extensive hydrophobic contacts shared among both classes of compounds. Moreover, the A ring moiety of cyclostreptin (Figure 1) occupies a space that is otherwise occupied by the βArg278 side chain. These observations suggest that covalent cyclostreptin-binding to βHis229 restricts βH7 dynamicity and sterically precludes M-loop structuring in a similar fashion as observed for the group II compounds.

### 2.3. Assembly Promoting Activity of Covalent Taxane-Site Binding Agents

Our cyclostreptin structure indicated that taxane-site ligands may favor microtubule assembly not only by assisting (at least partially, as for discodermolide, taccalonolide AJ and dictyostatin) folding of the M-loop in unassembled tubulin heterodimers in solution but also by other mechanisms. However, we also considered the possibility that the differences observed between diverse ligands with regard to folding of the M-loop could be related to differences in the assembly-promoting activities of these compounds. Thus, we measured the assembly-promoting activity of all three compounds that are known to bind covalently within the taxane-site (zampanolide, taccalonolide AJ, and cyclostreptin) by assaying tubulin polymerization under conditions where microtubule assembly does not occur on its own (phosphate buffer without glycerol) [23] (Figure 5). Under these conditions assembly and ligand binding are tightly linked and only ligated protein can assemble, thus allowing a direct determination of the elongation constant for the addition of a ligated dimer to the microtubule end (K_el_) [24], which is the inverse of the critical concentration of tubulin required for assembly [25]. Since taxane-site ligand-induced microtubule assembly is Mg^2+^ dependent [24], we also evaluated both low (3 mM) and high (7 mM) Mg^2+^ concentrations [23,24]. We found that at low Mg^+2^ concentrations (Figure 5A) only zampanolide (red line) induced microtubule assembly, whereas cyclostreptin (blue line) and taccalonolide AJ (green line)-treated tubulin remained unassembled (no difference to the vehicle-treated controls) (black line). At high Mg^2+^ concentrations (Figure 5B) all three compounds induced tubulin assembly. However, the increase of the signal (which is related to the number and length of microtubules assembled) and the lag time (which reflects to the nucleation step during initial microtubule polymerization) were different. Zampanolide (red line) produced a strong initial rate of microtubule assembly, which correlates with its ability to rapidly induce tubulin assembly also at lower Mg^2+^ concentrations. Consistent with previous reports, taccalonolide AJ-induced assembly (green line) displayed a lag time of ~20 min [18] with a maximal signal corresponding to approximately half of that for zampanolide. Cyclostreptin (blue line) demonstrated a polymerization lag of >20 min and a maximum signal half that of taccalonolide AJ.

These results closely correlate the M-loop stabilization states found in zampanolide and cyclostreptin structures with the assembly promoting activity of these compounds. Importantly, the DDM-paclitaxel hybrid (KS-1-199-132), which interacts with and stabilizes the M-loop (5LXS) also displays a higher assembly promotion activity than discodermolide despite a lower binding affinity [17].

### 2.4. Potency of Covalent Binders in βIII-Tubulin Expressing Paclitaxel-Resistant Cells

It is well established that, in addition to P-glycoprotein overexpression, βIII-tubulin isotype expression is a main clinical determinant of resistance to tubulin-targeted therapy [26,27,28]. Therefore, we evaluated the potency of the covalent tubulin binding agents to inhibit the growth of βIII-tubulin overexpressing HeLa cells as compared to isogenic controls to determine whether covalent binding was sufficient to circumvent this mechanism of resistance (Table 1).

As expected [7], βIII-tubulin expressing HeLa cells were resistant to paclitaxel with a resistance index of 16.6 as compared to the isogenic parental line (Table 1). While these cells were less resistant to cyclostreptin, taccalonolide AJ, and zampanolide with resistance indices of 2.8, 1.6, and 4.9 respectively, cells overexpressing βIII-tubulin were still somewhat resistant to these covalent tubulin binding agents. Conversely, pironetin, a covalent binder that targets the interdimer interface resulting in microtubule destabilization, is able to fully overcome this resistance with a resistance index of 0.6, suggesting that cells overexpressing βIII-tubulin were actually more sensitive to the drug than the parental line. These results suggest that covalent binding to tubulin facilitates overcoming βIII-tubulin-mediated drug resistance but with a greater impact when binding to the interdimer interface to destabilize microtubules than at the taxane-site to stabilize microtubules.

## 3. Discussion

### 3.1. Cyclostreptin Interaction with βHis229 of the Taxane-Site Induces Tubulin Assembly Without Structuration of the M-Loop

A detailed understanding of the structural and thermodynamic mechanisms employed by covalent microtubule stabilizers is crucial in order to design and develop compounds with optimal properties. The thermodynamic properties of the binding of non-covalent microtubule stabilizers are based on the difference in affinity of the ligand between the unassembled tubulin heterodimer K_bin1_ (Scheme 1) and the assembled microtubule K_bin2_. Compounds that have higher affinity for the polymerized state than for the unassembled curved state K_bin2_>>K_bin1_ displace the assembly equilibrium towards the right part of the reaction by decreasing the free concentration of unliganded microtubules [Mtb_+1_] towards the liganded assembled species [Mtb_+1_-Lig] [24,29] (Scheme 1).

The structural reasons for this difference in binding affinity for the dimer and the microtubule are not yet clear. In the case of the epothilones, which induce a structural change in the M-loop, the differences in binding affinity might arise from the difference in the M-loop structure between the straight assembled form, in which the loop is structured [21] and the curved unbound form, in which the loop is unstructured [20]. When an epothilone binds to unassembled tubulin, part of the energy of binding has to be employed to structure the M-loop; this is not required in the assembled form where the M-loop is already structured. Therefore, the free energy of binding to the assembled microtubule with the structured M-loop is lower than binding to the unassembled tubulin heterodimer with an unstructured M-loop. However, dictyostatin or discodermolide binding within the taxane-site promotes microtubule stabilization without fully structuring the M-loop [16,17], demonstrating a lack of a full understanding of the differences in affinity of taxane-site drugs between the assembled and unassembled forms of tubulin.

In the case of compounds that covalently bind to the taxane-site (zampanolide, cyclostreptin, and the taccalonolides) there is no difference in affinity for the unassembled and the assembled forms because the binding reaction is irreversible. In this case, the assembly induction should arise from a higher affinity (K_el2_) of the adducted protein for the microtubule end as compared to the non-adducted protein (K_el1_) (Scheme 2). Since 1/K_el_ is equal to the critical concentration for tubulin assembly [24,25], the increase in binding affinity results in a decrease in the tubulin concentration required for assembly.

Previous structural studies have shown that M-loop structuring is promoted by ligands with their respective side chains capable of establishing extensive interactions with the M-loop, as observed in the tubulin-complexes of zampanolide and epothilone [14] or the DDM-paclitaxel hybrid KS-1-199-32 [17]. Accordingly, since the M-loop is structured in the assembled form, the energy required to structure the M-loop in the non-adducted form has to be subtracted from the energy provided by the tubulin dimer addition to the microtubule end, which would not be the case if the M-loop is structured (in the adduct). This implies that the free energy of binding of the adduct to the microtubule end will be lower, and thus the adduct would be more prone to assembly than the non- modified protein Kel2>>Kel1. The observation that the DDM-paclitaxel hybrid KS-1-199-32, which in contrast to DDM, stabilizes the M-loop into a helical conformation, has a lower affinity for microtubules than DDM itself, but has more assembly promotion activity [17], further supports the hypothesis that structuration of the M-loop is relevant for promotion of assembly.

It should be mentioned that in chain D of the presented cyclostreptin complex the βT5-loop adopts the conformation observed for the GTP bound state despite of having GDP bound at the E-site. A closer inspection of the electron densities defining the βT5-loop conformations present in all available crystal structures of taxane-site ligand-tubulin complexes reveals a preference for the GTP state, with a few structures featuring both conformational states of the βT5-loop (Zampanolide, Epothilone A). One exception is dictyostatin, one of the best microtubule stabilizers having a moderate tubulin-binding affinity [16,23]. In the tubulin-dictyostatin complex the M-loop is not fully structured and the βT5 loop is in the GDP state. These observations suggest a potential implication of the βT5-loop in a common mechanism of tubulin activation for assembly promoted by taxane-site ligands, however, more detailed structural information is needed to allow a conclusive statement to support this hypothesis.

Our high-resolution structure of cyclostreptin-bound tubulin together with new MS-HPLC experiments performed at a higher resolution challenge our previous hypothesis of cyclostreptin binding to βAsn228 and βThr220 [6] and instead demonstrates that cyclostreptin unequivocally interacts with βHis229, a residue which is known to be responsible for the lack of sensitivity of yeast tubulin to paclitaxel [30]. In fact, given the key role of βAsn228 in stabilizing the guanine nucleotide at the E-site with two hydrogen bonds, a cyclostreptin interaction with this residue would severely affect nucleotide binding, which is not the case [31]. There is also no evidence supporting the βThr220 adduct we observed previously [6]. However, we cannot discard possible cyclation events in the mass analyzer during the fragmentation that prevents its detection. Considering a lack of extra density in the region of βThr220 in the crystal structure, we suggest that, if this adduct exists, it is in a proportionally low or brief transient state.

In spite of the fact that cyclostreptin lacks the ability to stabilize the helical conformation of the M-loop, it weakly induces tubulin assembly, supporting its function as a microtubule stabilizing agent likely through stabilization of the βT5 loop in the GTP conformation. Although the results support the relevance of progressive M-loop structuration in the potency of tubulin assembly induction as proposed previously [14], this M-loop stabilization is not essential for drug-induced microtubule polymerization by taxane-site ligands. Stabilization of longitudinal interactions is also likely to result in a stabilizing effect as previously proposed [15].

### 3.2. Cytotoxicity of Tubulin Covalent Binders and Effect on Cancer Cell Resistance

The two main clinically relevant mechanisms of drug resistance of tubulin-targeting agents are expression of drug efflux pumps (P-glycoprotein, MDR-1, BCRP) belonging to the ATP-binding cassette family of proteins [32,33], which results in multidrug resistance, and expression of the βIII-tubulin isotype [26,28,34,35,36,37]. It is known that covalent tubulin binding agents circumvent resistance due to drug efflux pumps [6,7,8,38] but the effect of covalent binding on resistance to βIII-tubulin isotype expression has not yet been studied in depth.

It has been proposed that the resistance of tubulin-stabilizing chemotherapeutics in βIII-tubulin expressing cells arises from defective binding to the taxane-site of βIII-tubulin [39,40,41]. On the other hand, Kavallaris et al. [35] proposed that it is highly unlikely that the mechanism of resistance is a direct result of the effects of taxanes on βIII-tubulin. Instead, it has been suggested that βIII-tubulin has a role as a cellular survival factor, since it is expressed under conditions of cellular stress, such as hypoxia [42] or glucose deprivation [43], and regions of ischemic necrosis in glioblastoma tumors express high levels of this isoform [44]. Finally, the Jordan group [45] proposed that the resistance is because βIII-tubulin incorporation makes microtubules less responsive to the suppressive effects of paclitaxel, thus allowing cells to maintain sufficiently rapid microtubule dynamics even in the presence of paclitaxel.

The use of covalent binders to investigate the mechanism of βIII-tubulin-mediated resistance has the advantage of eliminating the parameter of binding affinity. In principle, when studying a covalent binding agent, the rate and extent of drug binding will not be affected by a mutation in the binding site as long as it is able to bind with fast on rate. As an example, zampanolide has been found to be active in cells with paclitaxel/epothilone and laulimalide/peloruside resistance mutations and attempts to generate a zampanolide resistant cell line by mutagenesis have so far been unsuccessful [46]. Considering that the rate of binding of compounds to the taxane-site is very fast, [47,48,49,50], it would be expected that differences in the rate of binding would be irrelevant to potency in long-term antiproliferative assays. Zampanolide is the most potent of the three covalent microtubule stabilizers with an IC_50_ in HeLa cells of 0.045 nM. Taccalonolide AJ has an IC_50_ of 6.2 nM in this cell line with cyclostreptin having an IC_50_ of 19 nM, suggesting a direct relationship between the promotion of purified tubulin assembly and potency for inhibition cellular proliferation, as previously proposed [23].

When we assessed the antiproliferative potency of covalent microtubule stabilizers in HeLa cells overexpressing the βIII-tubulin isotype, we found that they were less susceptible to this form of resistance than paclitaxel. The finding that irreversibly binding ligands are less susceptible to βIII-tubulin-mediated resistance than reversible ligands, demonstrates that binding has a role in βIII-tubulin resistance. However, it is important to note that each of the covalent compounds still demonstrated partial (2–5-fold) resistance to the βIII-tubulin overexpressing line as compared to the parental line. The inability to completely overcome this mechanism of resistance through covalent binding supports the binding-independent roles of βIII-tubulin in drug resistance described above. Importantly, the use of the covalent microtubule destabilizer pironetin allowed us to distinguish between the proposed mechanisms of βIII-tubulin as a survival factor and as a mediator of microtubule dynamics. Pironetin completely overcomes βIII-tubulin-mediated resistance and is actually 1.7-fold more potent in the βIII-tubulin overexpressing line than in the parental cell line. Since pironetin does not bind to the taxane-site it would be expected to be completely insensitive to any change in the binding site. If the role of βIII-tubulin in the resistance is as a cell survival factor we would anticipate that both covalent stabilizers and destabilizers would be similarly impacted by βIII-tubulin overexpression, which is not the case. The fact that βIII-tubulin overexpressing cells are more sensitive to pironetin than the parental cells is instead consistent with a mechanism of resistance that involves counteracting microtubule stabilization. Therefore, as suggested previously [51], we propose that βIII-tubulin mediated resistance to the taxanes occurs both through decreased drug binding as well as by counteracting their effects on microtubule stability.

## 4. Materials and Methods

### 4.1. Proteins and Chemicals

Purified calf brain tubulin, and chemicals were as described [52]. The stathmin-like domain of rat RB3 and the chicken TTL proteins preparations were done as described previously [53,54]. Cyclostreptin was synthesized as described [55]. Zampanolide was synthesized as described [8]. Taccalonolide AJ was prepared as described [18]. Pironetin was kindly provided by Juan Murga (UJI).

### 4.2. Cell Culture

An isogenic couple of HeLa S3 and HeLa βIII [56] cells were selected to investigate the influence of βIII-tubulin overexpression in the sensitivity to covalent tubulin binders. These cells were selected to allow a direct comparison of the results to previously published studies [16,57,58]. They were cultured in Dulbecco Modified Eagle Medium supplemented with 10% fetal calf serum, glutamine (2 mM), gentamycin (40 µg/mL), penicillin (100 IU/mL), and streptomycin (100 µg/mL). The medium for HeLa βIII cells was additionally supplemented with geneticin (0.5 mg/mL). Antiproliferation assays were performed as described [23]. The statistical significance of differences in IC_50_ values were evaluated using the t-test option implemented in the Sigma Plot 13 software package (version 13, Systat Software, Inc., San Jose, CA, USA).

### 4.3. Time Course of Binding of Cyclostreptin to Dimeric Tubulin

Samples containing 20 μM tubulin and 25 μM cyclostreptin were incubated at 25 °C in PEDTA buffer (10 mM NaPi, 1 mM EDTA and 1.5 mM MgCl_2_, pH 7.0) plus 0.1 mM GTP. Samples were taken at different times (0′, 30′, 1 h, 2 h, and 4 h) and extracted three times with 1 *v*/*v* of dichloromethane. Samples without tubulin were used as a control. Ligand was quantified by HPLC-MS as described [6]. A sample of the complex at 4 h was stained with uranyl acetate and visualized by electron microscopy as described [59] to confirm that tubulin protein remained unpolymerized in these conditions. The integrity of the protein was confirmed by checking the assembly capacity of the protein at different times by warming up samples taken at each time to 37 °C for 30′. Samples were then centrifuged at 50,000 rpm for 20 min and amount of pelleted protein was measured spectrofluorometrically at λexc 285nm and λems 323 nm.

### 4.4. Tubulin Assembly in the Presence of the Drugs

Samples of 25 μM tubulin in 10 mM sodium phosphate, 1 mM EDTA, 1 mM GTP, pH 7.0 were supplemented with either 3 mM or 7 mM MgCl_2_, final pH 6.7. Then 27.5 μM zampanolide, taccalonolide AJ, cyclostreptin or DMSO (vehicle) were added. The samples were warmed to 37 °C and the assembly of tubulin followed by turbidimetricaly at 350 nm in a Thermo Scientific Multiskan Sky plate reader.

### 4.5. Tubulin-Cyclostreptin Adduct Formation, Crystallization, Data Collection, and Structure Determination

Two approaches were performed to form the tubulin-cyclostreptin adduct for crystallization. In the first approach, the adduct was prepared by incubating 20 µM tubulin with 80 µM cyclostreptin in PEDTA 1.5 mM MgCl_2_ pH 7.0 buffer for 4 h at 25 °C. To prepare the T_2_R-TTL-cyclostreptin complex, which contains two tubulin heterodimers, one TTL and one RB3 molecule, the proteins were mixed and concentrated (Amicon MWCO 10) at 4 °C to a final complex concentration of 20 mg/mL.

In the second approach, microtubules were first assembled from 20 µM tubulin in 3.4 M glycerol, 10 mM sodium phosphate, 1 mM EGTA, 6 mM MgCl_2_, and 0.1 mM GTP pH 6.5 for 30 min, and subsequently incubated with 80 µM cyclostreptin for 3 h at 37 °C. Microtubules were then disassembled by the addition of 4.5 mM EDTA and RB3 and further incubated at 4 °C for 1 h. Finally, TTL was added and the complex was concentrated as described above.

Both the T_2_R-TTL-cyclostreptin samples were supplemented with 10 mM DTT, 0.1 mM GDP and 1 mM Alpha,beta-Methyleneadenosine triphosphate (AMPCPP) before setting up crystallization experiments by the sitting-drop vapor diffusion technique at 20 °C. Considering known conditions from previous structures [10,14,16,60,61] we initially screened with a gradient of 0–7% PEG 4000 and 0–11% glycerol and further optimized the conditions. The best crystals grew in 6% glycerol, 3% PEG4000, 0.1 M MES/imidazole pH 6.7, 30 mM CaCl_2_, 30 mM MgCl_2_, and 5 mM L-tyrosine.

For data collection, crystals were collected directly from the drop and subsequently transferred into reservoir supplemented with 16 and 20% glycerol in two steps. X-ray diffraction data were collected at the beamline X06DA of the Swiss Light Source (Paul Scherrer Institut, Villigen PSI, Switzerland). Images were indexed and processed using XDS [62]. Structure solution using the difference Fourier method (template model PDB ID 5LXT) and refinement were performed using PHENIX [63]. Model building was carried out iteratively using the Coot software [64]. Data collection and refinement statistics are given in the Appendix A. All the figures were prepared with PyMOL (The PyMOL Molecular Graphics System, Version 2.0, Schödinger LLC, New York, NY, USA, 2017).

### 4.6. Mass Spectrometry Analysis

Tubulin samples were digested with modified porcine trypsin (Promega) at a final ratio of 1:50 (trypsin-protein). Digestion proceeded overnight at 37 °C. After digestion, samples were vacuum-dried and finally dissolved in 1% acetic acid for LC-MS analysis. The resulting peptide mixtures were subjected to nano-liquid chromatography coupled to mass spectrometry for protein identification. Peptides were injected onto a C-18 reversed phase nano-column (75 µM I.D. and 50 cm, Acclaim PepMap, Thermo Fisher Scientific, Waltham, MA, USA) and analyzed in a continuous acetonitrile gradient consisting of 0–40% B in 120 min, 50–90% B in 1 min (B= acetonitrile, 0.1% formic acid). A flow rate of 200 nL/min was used to elute peptides from the RP nano-column to an emitter nanospray needle for real time ionization and peptide fragmentation on a Q Exactive HF mass spectrometer (Thermo Fisher Scientific). One survey full scan and the 15 most intense fragmentation spectra were analyzed along the chromatographic run. Dynamic exclusion was set at 30 s. For protein identification, tandem mass spectra were extracted and charge state deconvoluted by Proteome Discoverer 1.4.0.288 (Thermo Fisher Scientific). All MS/MS samples were analyzed using SEQUESTTM (Thermo Fisher Scientific). Sequest was searched with a fragment ion mass tolerance of 30 ppm and a parent ion tolerance of 15 ppm. Oxidation of methionine was specified in Sequest as a variable modification. Custom application in R (Vseq) was developed to calculate and generate quality control graphs to monitor the specificity in peptide identification [65].

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
