# Peer review of "Crystal Structure of the Cyclostreptin-Tubulin Adduct: Implications for Tubulin Activation by Taxane-Site Ligands"

_ijms, 2019, doi:10.3390/ijms20061392_

Round 1

Reviewer 1 Report

 The article “Crystal structure of the cyclostreptin-tubulin adduct: Implications for tubulin activation by taxane-site ligands” is potentially interesting and well prepared.

The "Introduction" section is short, but it introduces the topic of work well.

The "Results" section -  the experiments are well performed and controlled, and the data appropriately interpreted, but in my opinion additional analysis with mutation of  beta-His229, it would support the results posted.

The "Materials and Methods" section is generally extensively described
The discussion section is too short, but I appreciate the brevity of the authors' statements. The authors summarized the results and compared them with the research of other teams. The conclusions drawn from the data analysis are correct. All results have been discussed, so I do not consider this a critical shortcoming.

Author Response

 The article “Crystal structure of the cyclostreptin-tubulin adduct: Implications for tubulin activation by taxane-site ligands” is potentially interesting and well prepared.

The "Introduction" section is short, but it introduces the topic of work well.

The "Results" section -  the experiments are well performed and controlled, and the data appropriately interpreted, but in my opinion additional analysis with mutation of  beta-His229, it would support the results posted.

The "Materials and Methods" section is generally extensively described

The discussion section is too short, but I appreciate the brevity of the authors' statements. The authors summarized the results and compared them with the research of other teams. The conclusions drawn from the data analysis are correct. All results have been discussed, so I do not consider this a critical shortcoming.

We agree with the reviewer, a mutation at bHis229 showing lack of interaction with cyclostreptin would undoubtly agree with the results described.

However, we would maintain that such additional experiments, which would be indirect in nature, are not critical to support these conclusions since we have obtained direct evidence of the reactivity of the residue using a well-established structural system (X-ray crystallography) and also high resolution mass spectrometry.

On the other hand, tubulin is a eukaryotic protein with an intricate folding pathway where different molecular chaperones are involved. Prefoldin and cytosolic chaperonin containing TCP1/TriC are required in the first part of the pathway, while tubulin folding cofactors (TBCs) are involved in the formation of the α‐ and β‐tubulin heterodimers. It means that the production of such mutant cannot be easily achieved in a standard prokaryotic system while production in a eukaryotic system cannot be done in a short time, the 10 days we were limited to by the editors.

Finally, it has been demonstrated that yeast tubulin, which incorporates asparagine in position b229 is insensitive to paclitaxel, while sensitivity is restored when this residue is mutated to histidine  (Gupta, et al., 2003). This is now mentioned in the text.

Gupta, M.L., Jr., Bode, C.J., Georg, G.I., and Himes, R.H. (2003). Understanding tubulin-Taxol interactions: mutations that impart Taxol binding to yeast tubulin. Proc Natl Acad Sci U S A 100, 6394-6397.

Reviewer 2 Report

The authors performed multidisciplinary approach to understand how tubulin assembles into stable microtubules as a result of the binding small molecules to the taxane-site. Most contents have been well-organized. However, this reviewer would like to suggest some additional things to describe in biological approaches.

1. The authors should describe the purpose of study using HELA cell lines only despite they indicated that it can target most solid tumors or hematological malignancies. Paclitaxel is used for standard regimen to treat ovarian, breast, lung, bladder, and prostate cancers.

2. Although the authors claim to be able to overcome paclitaxel resistance by class III β-tubulin overexpression, there will be several challenges to select the cancer patients with class III β-tubulin overexpression for treatment specifically. If so, the authors' discovery will also be challenging for the application of paclitaxel in general chemotherapy to treat various cancers. This points raised by this reviewer must be discussed.

Author Response

Comments and Suggestions for Authors

The authors performed multidisciplinary approach to understand how tubulin assembles into stable microtubules as a result of the binding small molecules to the taxane-site. Most contents have been well-organized. However, this reviewer would like to suggest some additional things to describe in biological approaches.

1.       The authors should describe the purpose of study using HELA cell lines only despite they indicated that it can target most solid tumors or hematological malignancies. Paclitaxel is used for standard regimen to treat ovarian, breast, lung, bladder, and prostate cancers.

The reason for using HeLa cells is to allow comparisons with results previously obtained in two standard sets of isogenic cell lines that show resistance to paclitaxel chemotherapy by different mechanisms.

Set 1 comprises HeLa, HeLa p-glycoprotein overexpressing and HeLa overexpressing βIII tubulin. (Tang, et al., 2017)

Set 2 comprises A2780, A2780 p-glycoprotein overexpressing and A2780 carrying mutations in the beta tubulin gene. (Trigili, et al., 2016)

We have collected these two sets using stable cell lines from different sources with the intent to compare the efficacy of the compounds in both sets of isogenic cell lines.

We have tested a large number of compounds in these systems. Testing new compounds in a different system would prevent us from correlating new data with those from our previous studies on other compounds.

We now mention in the M&M section that the set of cells was selected so we could compare the results of this study with these of previous works.

2. Although the authors claim to be able to overcome paclitaxel resistance by class III β-tubulin overexpression, there will be several challenges to select the cancer patients with class III β-tubulin overexpression for treatment specifically. If so, the authors' discovery will also be challenging for the application of paclitaxel in general chemotherapy to treat various cancers. This points raised by this reviewer must be discussed.

We appreciate the comment, but we are far from suggesting that we have found a way to treat cancer patients with Class III β-tubulin overexpression. The only intention of our work is to use a tool (covalent binders) to understand the not yet understood molecular mechanisms of Class III β-tubulin-mediated resistance. We hope the results and conclusions presented here can be of use for clinical researchers that could exploit our findings in the search for improved treatments.

Tang, Y., Rodriguez-Salarichs, J., Zhao, Y., Cai, P., Estevez-Gallego, J., Balaguer-Perez, F., Redondo Horcajo, M., Lucena-Agell, D., Barasoain, I., Diaz, J.F., et al. (2017). Modification of C-seco taxoids through ring tethering and substituent replacement leading to effective agents against tumor drug resistance mediated by betaIII-Tubulin and P-glycoprotein (P-gp) overexpressions. Eur J Med Chem 137, 488-503.

Trigili, C., Barasoain, I., Sánchez-Murcia, P.A., Bargsten, K., Redondo-Horcajo, M., Nogales, A., Gardner, N.M., Meyer, A., Naylor, G.J., Gómez-Rubio, E., et al. (2016). Structural Determinants of the Dictyostatin Chemotype for Tubulin Binding Affinity and Antitumor Activity Against Taxane- and Epothilone-Resistant Cancer Cells. ACS Omega 1, 1192-1204.

Round 2

Reviewer 2 Report

So, this reviewer accept that the authors do not want deepening discussion with the current research stage. Nevertheless, the final goal of the authors' work should be to keep the clinical application in mind.